# Research on GA-SVM Based Head-Motion Classification via Mechanomyography Feature Analysis

**DOI:** 10.3390/s19091986

**Published:** 2019-04-28

**Authors:** Yue Zhang, Jing Yu, Chunming Xia, Ke Yang, Heng Cao, Qing Wu

**Affiliations:** Department of Mechanical Engineering, East China University of Science and Technology, Shanghai 200237, China; y10180239@mail.ecust.edu.cn (Y.Z.); shelly_yujing@163.com (J.Y.); thefuturehere@163.com (K.Y.); hengcao@163.com (H.C.); qwu@ecust.edu.cn (Q.W.)

**Keywords:** mechanomyography, genetic algorithm, support vector machine, head-motion, classification

## Abstract

This study investigated classification of six types of head motions using mechanomyography (MMG) signals. An unequal segmenting algorithm was adopted to segment the MMG signals generated by head motions. Three types of features (time domain, time-frequency domain and nonlinear dynamics) were extracted to construct five feature sets as candidate datasets for classification analysis. Genetic algorithm optimized support vector machine (GA-SVM) was used to classify the MMG signals. Three different kernel functions, different combinations of feature sets, different number of signal channels and training samples were selected for comparative analysis to evaluate the classification accuracy. Experimental results showed that the classifier had the best overall classification accuracy when using the radial basis function (RBF). Any combination of three different types of feature sets guaranteed an average accuracy of over 80%. In the case of the best combination (feature set 2 + 3 + 5), the classification accuracy was up to 88.2%. Using four channels to acquire MMG signal and no less than 60 training samples can assure a satisfactory classification accuracy.

## 1. Introduction

Recent research activities about classification of human body movements are mainly focused on the classification for limb-motions, more frequently using surface electromyography (sEMG), which is a sort of bioelectric signal of neuromuscular system that is recorded from the skin surface. Cheng et al. [1] adopted a discriminant bispectrum feature extraction approach based on sEMG, and used support vector machine (SVM) to classify nine types of hand and wrist motions. Oskoei et al. [2] demonstrated that SVM has better performance than linear discriminant analysis (LDA) and multilayer perceptron (MLP) in the investigation of classifying upper limb motions. The sEMG-based recognition of upper limb motions has been applied in the control system of rehabilitation equipment and provide objective data for quantitative assessment [3]. In particular, the pattern recognition of hand motion was applied to real-time control of sEMG-based multifunctional prosthesis [4]. The classification of individual and combined finger movements was adopted to control the finger postures of a prosthetic hand [5]. The classification of hand signs based on data collected from accelerometers, and sEMG were used to identify sign language for hearing-impaired and non-verbal community [6]. Kuang et al. [7] proposed a recognition method based on extreme learning machine (ELM) to recognize the patterns of seven lower limb movements, with the overall recognition accuracy above 95%.

Mechanomyography (MMG), a superficial measurement of mechanical vibrations [8], has recently been used in the classification of human body movements. Alves et al. [9] selected fourteen features and used LDA on multi-channel MMG analysis to achieve hand movements recognition rate higher than 90%. Wu et al. [10] used SVM on MMG features for real-time continuous recognition of knee motion and acquired the average accuracy up to 91%. Ding et al. [11] extracted three feature sets including wavelet packet transform (WPT) coefficients, stationary wavelet transform (SWT) coefficients, time and frequency domain hybrid (TFDH) features, adopted SVM for the classification of five finger-motions, and obtained the accuracy up to 91%. These research results verified the feasibility to conduct pattern recognition of body movements based on MMG feature analysis. However, further investigations need to be carried on, for instance, how to combine time domain, time-frequency domain and nonlinear dynamics features properly, how to optimize SVM parameters, and how to determine the number of signal acquisition channels and the number of training samples. Some quadriplegics who clinically suffered from severe loss of motor function of limbs cannot manipulate some normal devices as normal people. However, they could execute head motions. For them, it is possible to manipulate some simple devices according to the recognition results of head motions, such as turning on or off the television and switching the channels, turning on or off the air-conditioner and regulating the temperature, as well as even controlling the electric wheelchair. This technology has the potential to improve their life quality. In addition, there is a lack of research on the classification of multi head-motion via sEMG or MMG analysis.

MMG is regarded as a mechanical counterpart of the electrical activation of skeletal muscle [12]. Before the acquisition of sEMG, it is necessary to perform the careful preparation of the skin (shaving, abrasion, and cleaning with alcohol) [13]. However, no skin preparation is required when conducting the MMG acquisition, thus it is convenient to perform experiments. In this paper, fifteen features were extracted from four-channel MMG signals corresponding to six types of head motions. These features were divided into five candidate sets for constructing feature vectors. Genetic algorithm-support vector machine (GA-SVM) method was adopted to classify six head motions, and comparisons were performed for different kernel functions, the selections of feature set combinations, the numbers of acquisition channels and the numbers of training samples. Finally, the effect factors of classification accuracy of head motions based on MMG signals are discussed.

## 2. Materials and Methods

### 2.1. Subjects

Eight healthy male students (age: 24.4 ± 0.9 years, height: 175.6 ± 4.0 cm, weight: 64.1 ± 5.8 kg) with no history of neuromuscular disease volunteered to participate in the study. They were fully informed of the content of the experiments and then signed the informed consent. All of them were not engaged in strenuous exercise 24 h before the experiments.

### 2.2. Experimental Protocol

All the subjects were instructed to perform six types of head motions: forwarding, backwarding, swinging to left, swinging to right, turning to left and turning to right, as shown in Figure 1. Each motion was done repeatedly 100 times every 3 s. To avoid muscle fatigue, each subject rested for 30 min after completing a type of motions.

MMG signals were measured by four accelerometers (TD-3, Beijing, China) on the surface of each side of sternocleidomastoids and splenius capitis and amplified by an amplifier with a gain of 2000. Then, the MMG signals were digitized by a 16-bit data acquisition card (NI-9205, Austin, TX, USA) with a sampling rate of 1000 Hz. The original MMG signals were filtered by an elliptic digital filter with the passband of 5–100 Hz. Data analysis was performed on a laptop (Inter Core i5 2.50 GHz, RAM 4.0 GB) with MATLAB (MathWorks Inc, Natick, MA, USA).

### 2.3. Signal Segmentation

Since the length of MMG signal corresponding to each motion is not equal, Jiang and Xia [14] proposed an unequal length segmentation algorithm. In the condition that the number of motions is known, the MMG signal segment of each motion can be obtained. The process of acquiring the corresponding signal length is as follows.

[Step 1]Obtaining the secondary envelopes of the pre-processed MMG waveform.[Step 2]Identifying all the maximum points and minimum points on the envelope line and sorting these points in sequence.[Step 3]Each maximum point has a head minimum and a rear minimum point. Cut out the signal frame between the two nearby minimum points. Thus, the two minimum points correspond to the start point and the end point. Accordingly, all the signal frames are acquired.[Step 4]Calculating the maximum absolute value (summit value) of each signal frame. The sequence value of each signal frame depends on the position of the point with summit value. All the signal frames are sorted as summit values from large to small (1 to N).[Step 5]Merging adjacent signal frames: comparing the Nth signal frames to the (N − 1)th one, if the start point of the Nth signal frame coincides with the end point of the (N − 1)th frame, merging the Nth signal frame with the (N − 1)th one and moving all the sequence values of all the ones behind N forward with 1; otherwise, comparing the Nth signal frame with the (N − 2)th one as above.[Step 6]Removing non-motional signal frames: sorting the N signal frames according to sequence numbers from small to large, then calculating the absolute values {D(n)} of the difference between each signal frame with the one behind. If the absolute value D(j) of the difference between the jth signal frame with the (j + 1)th one is much less than DA (the average value of {D(n)}), the smaller one is considered as a non-motional signal frame and removed, back to [Step 4]; otherwise the N signal frames were considered as all the motional signal frames.

After these steps, the MMG signal was divided into N motion frame signal segments. The start point and the end point of each frame signal in the sampling time series were obtained. According to the steps above, the MMG signals of each movement during the whole experiment were cut for each subject, as shown in Figure 2.

### 2.4. Feature Extraction

Root mean square (RMS), Variance (VAR), Zero crossing (ZC), modified mean absolute value (MMAV), waveform length (WL), and log detector (LOG) were selected as time domain features [15] as in Equations (1)–(6):(1)$$RMS=\sqrt{\frac{1}{N}\sum_{i=1}^{N}x_{i}^{2}}$$
where $x_{i}$ is the value at the ith sampling and N is the number of sampling points.
(2)$$VAR=\frac{1}{N-1}\sum_{i=1}^{N}x_{i}^{2}$$
(3)$$ZC=\sum_{j=1}^{N-1}\mathrm{sgn}(-x_{j}\cdot x_{j+1}),\;\mathrm{where}\;\mathrm{sgn}(x)=\left\{ \begin{array}{l} 1,\;\mathrm{if}\;x>0\\ 0,\;\mathrm{if}\;x\leq 0 \end{array}\right.$$
(4)$$MMAV=\frac{1}{N}\sum_{i=1}^{N}w_{i}\left|x_{i}\right|,\;\mathrm{where}\;w_{i}=\left\{ \begin{array}{ll} 1, & \;\mathrm{if}\;0.25N\leq i\leq 0.75N\\ 0.5, & \;\mathrm{otherwise} \end{array}\right.$$
(5)$$WL=\sum_{i=1}^{N-1}\left|x_{i+1}-x_{i}\right|$$
(6)$$LOG=e^{\frac{1}{N}\sum_{i=1}^{N}\log(\left|x_{i}\right|)}$$

The wavelet packet decomposition was applied to the time-frequency feature extraction. The subspace composed of the scale function ϕ(t) and the subspace $\left\{V_{j}\right\}$ composed of the wavelet function ψ(t) have a relationship of $V_{j}⊥W_{j}$ and $V_{j+1}=V_{j}⊕W_{j}$, $V_{j}$ continues to be decomposed, as shown in Equation (7).
(7)$$V_{j+1}=V_{j-1}⊕W_{j-1}⊕W_{j}=V_{j-2}⊕W_{j-2}⊕W_{j-1}⊕W_{j}=\cdots$$

According to the Equation (8)
(8)$$V_{j}=U_{j}^{0},\;W_{j}=U_{j}^{1},\;U_{j}^{2n}⊥U_{j}^{2n+1},\;U_{j+1}^{n}=U_{j}^{2n}⊕U_{j}^{2n+1}$$

Further decomposition
(9)$$\begin{array}{l} W_{j}=U_{j-1}^{2}⊕U_{j-1}^{3}=(U_{j-2}^{4}⊕U_{j-2}^{5})⊕(U_{j-2}^{6}⊕U_{j-2}^{7})\\ \cdots \\ =U_{j-k}^{2^{k}}⊕U_{j-k}^{2^{k}+1}⊕\cdots ⊕U_{j-k}^{2^{k+1}-1}\\ \cdots \\ =U_{0}^{2^{j}}⊕U_{0}^{2^{j}+1}⊕\cdots ⊕U_{j-k}^{2^{j+1}-1} \end{array}$$

The 6-scale wavelet packet decomposition was performed on the signal to obtain wavelet packet coefficients. According to Parseval theorem, Wavelet packet node [6,1] energy (WP [6,1]), WP [6,3], WP [6,2], WP [6,6], WP [6,7], WP [6,5] was calculated as time-frequency domain features. Approximate Entropy (ApEn), Fuzzy Entropy (FuzzyEn) and Sampling Entropy (SampEn) were calculated as nonlinear dynamic features. The details of the specific method refer to [16,17,18].

In order to investigate the effects obtained by using different features above, we divided the featuresby nature, and selected several sets for classifier training for comparison. All the features above were divided into five feature sets (codes 1–5). Feature set 1 includes commonly used time domain features; feature set 2 includes time domain feature [15], which can achieve a satisfactory classification accuracy but are not frequently used; feature set 3 includes relatively low-frequency time-frequency domain features; feature set 4 includes relatively high-frequency time-frequency domain features; and feature set 5 includes nonlinear dynamic features, as shown in Table 1. All the feature values were performed normalization to [0,1]. Based on the four channel signals, part of the five feature sets were selected to form a high-dimensional feature vector to train the classifier. In order to obtain a better classification, it was worth using more than one feature set, at the same time, the number of features and samples should satisfy the proportional relationship within a certain range [19]. In this study, two or three feature sets were used respectively.

### 2.5. Classification by GA-SVM

SVM is a machine learning method based on the statistic theory with excellent generalization ability and practicability. This method has unique advantages in pattern recognition of small training samples.

The model corresponding to the hyperplane in the feature space is
(10)$$y=w^{T}x+b$$
where x is the input set, y is the output set, $w^{T}$ is the normal vector that determines the direction of the hyperplane, and b is the offset.
(11)$$\underset{w,b}{\min}\frac{1}{2}{\left\|w\right\|}^{2}$$
subject to $y_{i}(w^{T}x_{i}+b)\geq 1,\;i=1,\cdots ,m$

An appropriate kernel function maps x to a higher dimensional feature space and solve the optimization problem:(12)$$\underset{\alpha }{\min}\frac{1}{2}\sum_{i=1,j=1}^{n}y_{i}y_{j}\alpha_{i}\alpha_{j}\kappa (x_{i},x_{j})-\sum_{i=1}^{n}\alpha_{j}$$
subject to $\sum_{i=1}^{l}y_{i}\alpha_{i}=0$, i=1,⋯,l

The optimal solution is $\alpha ={\left(\alpha_{1},\cdots ,\alpha_{l}\right)}^{T}$. Choose $\alpha_{j}$ to calculate the threshold:(13)$$b=y_{i}-\sum_{i=1}^{l}y_{i}\alpha_{i}\kappa (x_{i},x_{j})$$
construct the decision function:(14)$$f(x)=\mathrm{sgn}(\sum_{i=1}^{l}\alpha_{i}y_{i}\kappa (x_{i},x_{j})+b)$$

When SVM is applied in classification, both the error penalty parameter (c) and the kernel parameter (g) influence the performance of SVM. With the advantages of fitting large-scale parallel process and a better overall optimizing ability, GA is utilized to search the best parameters c and g in a definition domain for SVM, namely GA-SVM. GA-SVM has been applied in the classification of the muscle states (maximum voluntary contraction, fatigue degree) based on MMG signal. GA-SVM achieved a higher classification accuracy than the back-propagation neural networks (BP-NN) and SVM [20], the flowchart of GA-SVM could be seen in [21].

### 2.6. Training and Performance of the Classifier

K-fold Cross Validation (K-CV) is a common statistical analysis method to verify the performance of the classifier. It has been used in the evaluation of the performance of sEMG recognition system [22]. All samples for each type of motion were divided into K subsets, each of which was tested once, and the remaining K-1 subsets were adopted as training sets for GA-SVM. Comparing the classification results of testing samples with their true labels, the classification accuracy (CA) of all the testing samples are calculated as Equation (15).
(15)$$CA=\frac{N_{right}}{N_{all}}\times 100\%$$
where $N_{right}$ is the number of correctly classified samples, and $N_{all}$ is the number of all testing samples.

The CV accuracy is considered as the performance indicator of classifier performance (Equation (16)).
(16)$$CVA=\frac{1}{k}\sum_{j=1}^{k}A_{k}$$
where k is the number of folds, and $A_{k}$ is the accuracy measure of the $j_{th}$ fold.

In this study, K was 5, i.e., 20 of the 100 samples of each motion were used as a testing set, and the rest 80 samples were used for a training set. The searching ranges of c and g were in [0.1, 100] and [0.01, 10] respectively. Three kernel functions were tested, including radial basis function (RBF), linear and polynomial kernel. LibSVM3.14 toolbox [23] was used for programming implementation in this study. 

## 3. Results

### 3.1. Feature Sets Selection

The experimental result showed that the classification accuracies of classifier trained by single feature set were not satisfactory (the CV accuracies are lower than 80%, as shown in Figure 3). The highest accuracy was achieved by feature set 2, with classification accuracies of 79.2% (RBF), 78.1% (Linear) and 77.2% (Polynomial) respectively. While the lowest accuracy was using feature set 4, with the classification accuracies of 64.4% (RBF), 64.8% (Linear) and 58.2% (Polynomial) respectively. Therefore, when only one feature set was selected, the classification accuracies showed a certain difference.

Selecting different numbers of feature sets and different kernels of GA-SVM, the classification accuracies are shown in Figure 4 and Figure 5. When two feature sets were selected, the highest classification accuracy was achieved by feature sets (2 + 5), with accuracies of 86.3% (RBF), 86.6% (Linear) and 84.4% (Polynomial) respectively. The lowest was using feature set (3 + 4), which accuracies were 75.3% (RBF), 75.2% (Linear) and 69% (Polynomial) respectively. With three feature sets, the classification accuracies further increased, all above 80%. In particular, classification accuracies by feature set (2 + 3 + 5) were as high as 88.2% (RBF), 87.9% (Linear) and 85.5% (Polynomial) respectively. By comparing results of different kernel functions, RBF and Linear were better than Polynomial. In the following parts, RBF was used as the kernel function.

The accuracy of each motion was obtained by using three feature sets, as shown in Figure 6. When the feature sets consisted of the features of time domain, time frequency domain and nonlinear dynamics, the classification accuracy of each motion was higher than 80%.

To illustrate the overall result of classifying each sort of head movements, the confusion matrix was used as shown in Figure 7, where the column is the targeted motion and the row is the predicted motion. The results demonstrated that we could get a comparatively high classification accuracy for three types of head motions: ‘backwarding’, ‘swinging to the left’ and ‘swinging to the right’. The accuracy was comparatively low for ‘forwarding’, and misclassified to ‘turn to the right’ was up to 5.5%. The misclassification of ‘turning to the left’ into ‘turning to the right’ was up to 6.1%. Moreover, the misclassification of ‘turning to the right’ into ‘forwarding’ or ‘turning to the left’ was 4.1%.

### 3.2. Channels Selection

To figure out the influence of each channel on the classification accuracy, we selected two or three of the four channels to calculate the classification accuracies based on the combination of different channels. The result is shown in Figure 8.

The comparison results show that using three channels, the classification accuracy of channel 1&3&4 was the highest, reaching 84.4%, while the lowest using 1&2&4 was 81.7%. When using two channels, the classification accuracy using 3&4 was the highest, reaching 78.2%, while the lowest was only 71.5% by using channel 1&2. Therefore, the classification accuracy was influenced by the selected signal channels. Classification accuracy by using four channels was higher than using three channels or two channels, which embodies the rationality of using four channels of MMG signals in this study.

### 3.3. Numbers of Training Samples

To investigate the relationship between the classification accuracy and the number of training samples, using the method proposed in Section 2.6, twenty samples of each type of motions consisted of a testing set, while n (n = 60, 65, 70, 75, 80) of the rest 80 samples were used in the training stage, the CV accuracy is shown in Figure 9.

When the number of training samples increased from 60 to 80, there was a slight rise in the classification accuracy. When using the feature sets (2 + 3 + 5), (2 + 4 + 5) and (1 + 3 + 5), the classification accuracies increased from 86.8% to 88.1%, 86.0% to 87.4%, and 85.8% to 87.4% respectively. Even in the condition of less training samples (n = 60), the classification accuracy was still above 85%.

## 4. Discussion

In this study, we selected some features in time domain, time-frequency domain and nonlinear dynamics, which were frequently used in previous researches about sEMG or MMG. According to the results of this study, using single feature set 1, 2 or 5, i.e., time domain features and nonlinear dynamics features, could acquire better classification accuracy, nevertheless lower than 80%. When using two feature sets, the classification accuracy rose to a maximum of 86.3%. However, there was a great difference between different feature sets, e.g., using feature set (3 + 4) could obtain a low classification accuracy of 75.0%. When using three feature sets, the classification accuracy further improved, and all the classification accuracies exceeded 80%, with a maximum of 88.1%. Generally, selecting three feature sets, i.e., one time domain feature set (1 or 2) + one time-frequency feature set (3 or 4) and one nonlinear time series analysis feature set (5) could acquire reasonably good classification accuracy.

We tried to reduce the signal channels for the purpose of reducing sensors, thus two or three of the four channel signals were analyzed. The result revealed that the classification accuracy was more satisfactory when adopting four channel signals, though it was more expensive and less convenient. Whether more channel signals could further improve the classification accuracy requires additional investigation. Besides, whether we can select some other muscles, even in the case of fewer signal channels, could achieve higher classification accuracy is unknown.

MMG has been applied in the classification successfully for some motions recognition, indicating potential applications [24,25,26]. Comparing with some other studies, we adopted unequal segmenting algorithm to extract MMG signal of single motions rather than segmenting signals to a fixed length, thus being adaptive. From the result of comparing the numbers of training samples, the classification accuracy of testing samples improved over 1% as the number increased from 60 to 80. However, there is no evidence that the classification accuracy could be improved further as the number further increased.

Some physical challenged people such as spinal cord injury patients with high paraplegia have limited limb movements. They could use head motions instead of limb motions to control certain equipment to improve their living quality. The classification algorithm presented in this paper has potential application for these patients. Currently, this algorithm is only suitable for offline processing and the number of samples must be known. It is expected to be improved without knowing the exact number of samples and could be real-time processed. Some achievements have been made in the field of prosthetic research based on MMG signal. The discrimination in MMG signal of hand motions is large, which is easy to be recognized, leading to a high classification accuracy. However, the discrimination in MMG signals of head motion is small, which increases the difficulty of recognition, resulting in the relative lower classification accuracy. The classification of head motions based on MMG signals needs further investigation for higher classification accuracy and practical application.

## 5. Conclusions

This study presented comparative study of head-motion classification based on MMG. Our experiment has demonstrated that classification of six head motions is feasible by adopting GA-SVM algorithm based on extracted features of four-channel MMG. In the aspect of selecting kernel function, RBF is better for the classification. With respect to the selection of feature sets, using three feature sets enables the average classification accuracy higher than 80%; while using the combination of time domain feature, time-frequency domain feature and nonlinear dynamic analysis feature could get better classification accuracy, up to 88.2%. In terms of the number of channels, it is obvious that using four-channel signal gets better results than using three-channel or two-channel signal. The classification method is suitable for small training samples as it yields the satisfactory result when the number of training samples is between 60 to 80.

## Figures and Tables

**Figure 1 sensors-19-01986-f001:**
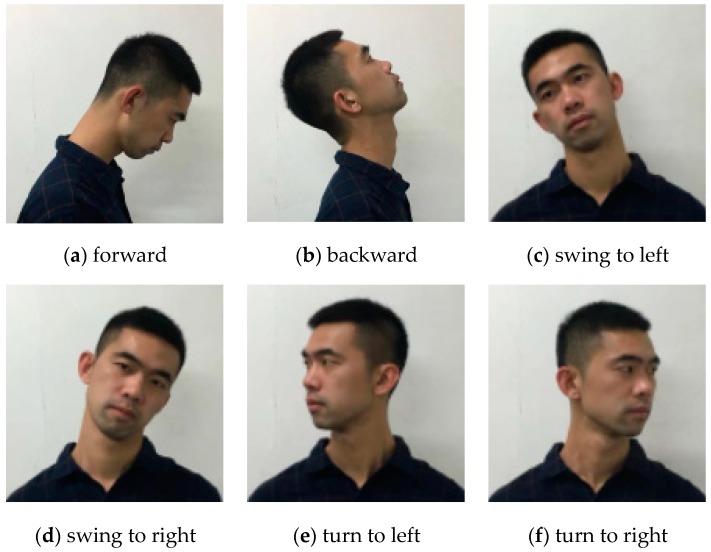
Six types of head motions.

**Figure 2 sensors-19-01986-f002:**
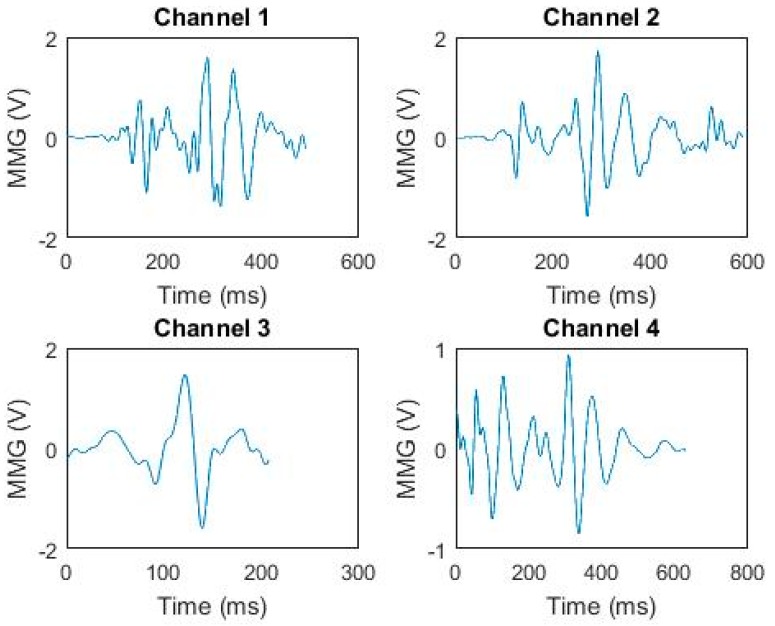
A typical segmentation of 4-channel pre-processed MMG signals.

**Figure 3 sensors-19-01986-f003:**
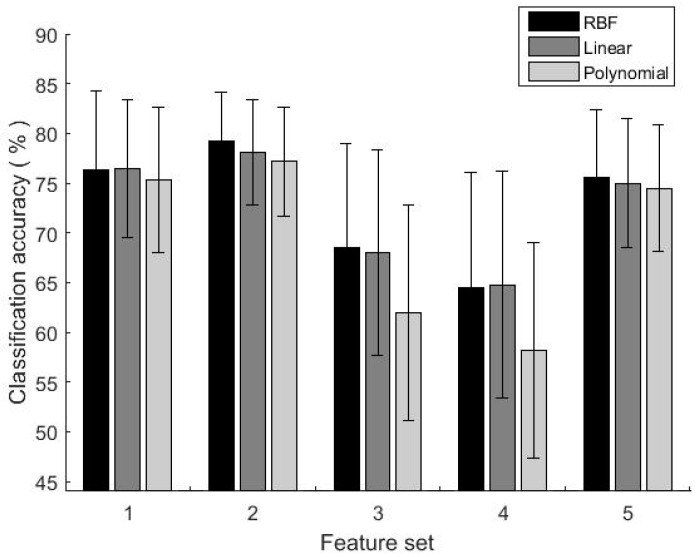
The classification accuracy using a single feature set.

**Figure 4 sensors-19-01986-f004:**
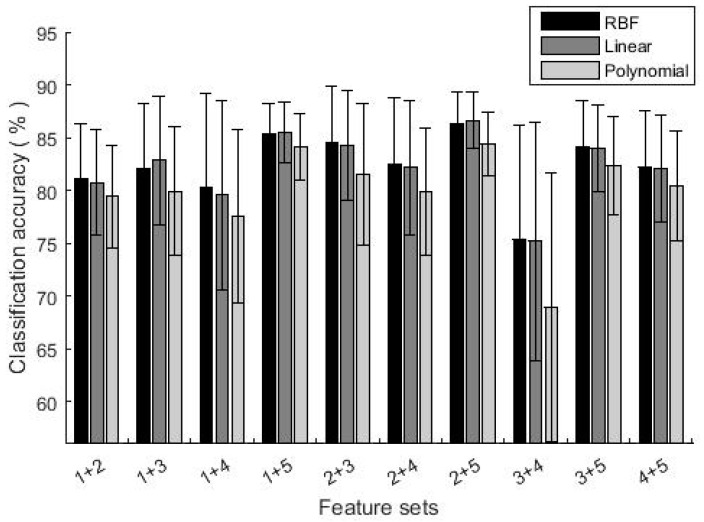
The classification accuracy using two feature sets.

**Figure 5 sensors-19-01986-f005:**
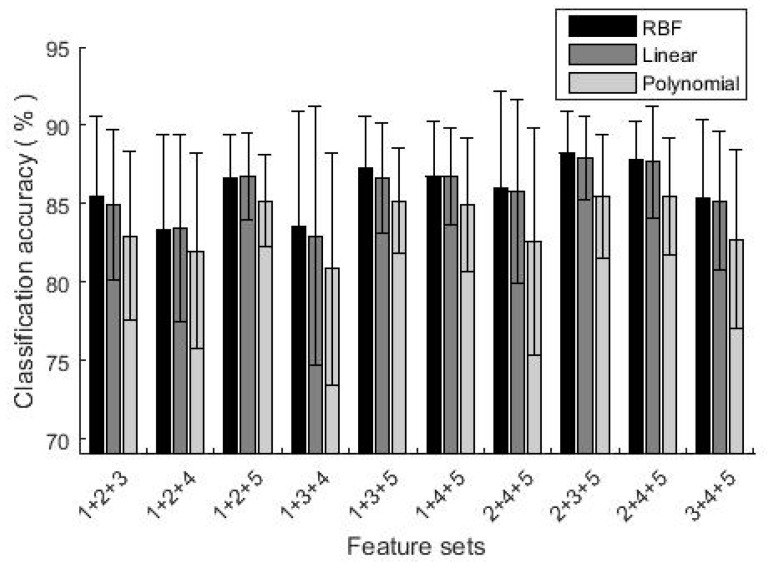
The classification accuracy using three feature sets.

**Figure 6 sensors-19-01986-f006:**
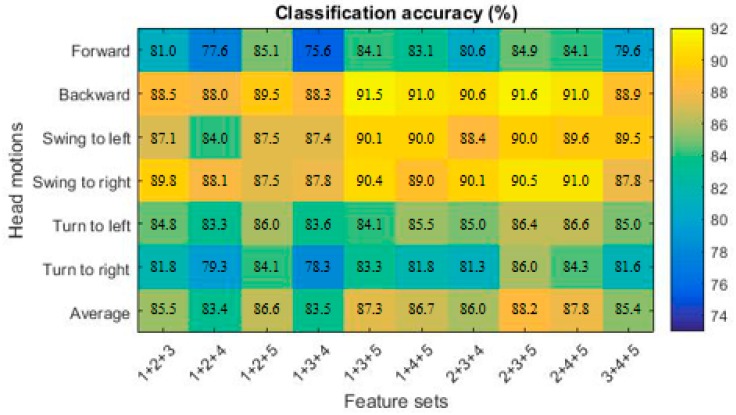
The classification accuracy of each head motion using three feature sets.

**Figure 7 sensors-19-01986-f007:**
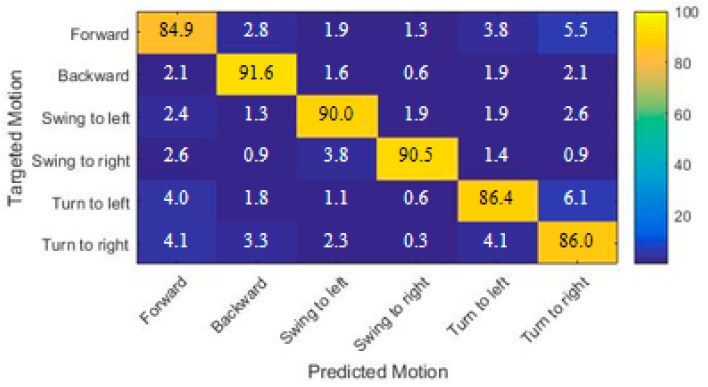
The confusion matrix of the classification (feature set (2 + 3 + 5), radial basis function).

**Figure 8 sensors-19-01986-f008:**
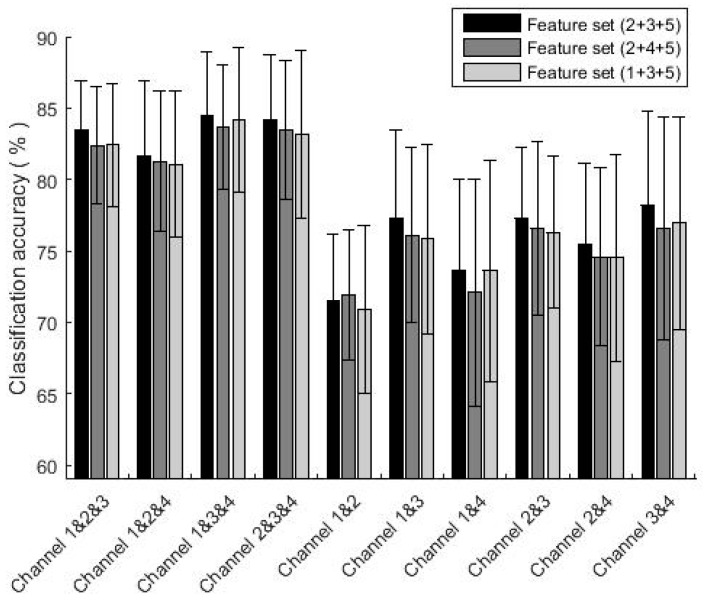
The classification accuracy of different feature sets based on the combinations of different channels.

**Figure 9 sensors-19-01986-f009:**
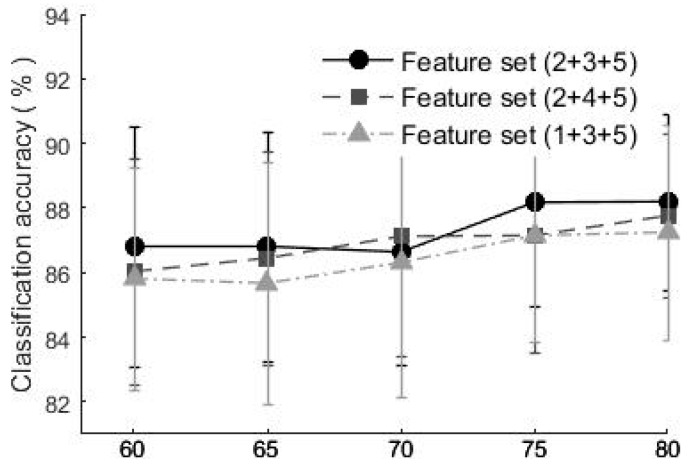
The classification accuracy based on different numbers of training samples.

**Table 1 sensors-19-01986-t001:** Feature set.

Vector Number	Feature Vector	Feature Type
feature set 1	RMS, VAR, ZC	Time domain
feature set 2	MMAV, WL, LOG	Time domain
feature set 3	WP (6,1), WP (6,3), WP (6,2)	Time-frequency domain
feature set 4	WP (6,6), WP (6,7), WP (6,5)	Time-frequency domain
feature set 5	ApEn, FuzzyEn, SampEn	Nonlinear dynamic

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
