# Peer review of "Research on GA-SVM Based Head-Motion Classification via Mechanomyography Feature Analysis"

_sensors, 2019, doi:10.3390/s19091986_

Round 1
Reviewer 1 Report
line 87, change "2000" to "2000x". One question, why a gain so high? generally we use 1x to MMG.
Congratulations for your work, the paper is so interesting, the methodology presents good results, however, your study was applied off line, I am curious to know what happens during a real time application. Sometimes, although we found great results in accuracy and others parameter in offline process, during the online (in practical) the results are different. Remember that the major goal in your work is improve the quality of life of other individuals.
Author Response
Thanks for your advices. The response to your comments could be seen in the attachment.

Reviewer 2 Report
This paper presents a head-motion classification using mechanomyography (MMG) feature analysis. The authors used four accelerometers attached to each side of sternocleidomastoids and splenius capitis muscles to measure the muscle vibration during six types of head motion. Some features were calculated to classify the head motions. The authors then showed classification performance of their suggested features. However, there is some issues to be addressed for the current manuscript in terms of the purpose of this study and methodology, as followings:
1. Aim of this study is not clear. For example, why do we need to classify the head motions?
2. Why did the authors use MMG? What if the acceleration data were collected from head directly?
3. Since acceleration data are extremely sensitive to motion artifact, MMG signals collected from muscles might not result from muscle vibration during six types of head motion that the authors defined. Probably, it might due to motion artifact and bulged muscle, rather than muscle vibration due to muscle contraction. This reviewer strongly suggest that the authors need to confirm what the primary source would be of acceleration signals. Also, general pattern of MMG signals during each head motion should be analyzed.
4. It seems that all four muscles chosen by the authors are necessary to classify six types of head motion. Reduction in the number of signal channel might not be appropriate in this study.
5. Sensitivity and specificity analysis should improve this paper. It may tell us how the algorithm would be practical.
6. What is the classification performance for each subject? The approach that the authors suggested is subject-specific?
Author Response
Thanks for your advices. The response to your comments and introduction for revision could be seen in the attachment.

Reviewer 3 Report
This manuscript regards with the head motion classification based on the genetic algorithm optimized support vector machine (GA-SVM). The authors have made a comparison among the classification accuracies by using different feature sets. In general, the manuscript really has certain academic value for the readers.
MAJOR REVISION
1. There are a lot of grammar errors in the present manuscript. Those errors should be corrected before the manuscript can be acceptecd for publication. Due to many errors, only some of them are listed in the minor revision.
MINOR REVISION
1. LINE 27
“focus” should be “focused.”
2. LINES 108~109
“And the two minimum …“ is not a complete sentence.
3. LINE 110
There are two words of ”the.”
4. LINE 124
“are’ should be “were.”
5. LINE 157
“… is to…” should be “…was to ….”
6. LINE 168
“… are…” should be “…were ….”
7. LINES 187 & 191
What does “s.t.“ mean in the two equations ? Clear description is needed for this shorthand.
8. LINE 198
“…the best parameter…“ should be “the best parameters….”
9. LINE 208
“…which is …“ should be “…which was….”
10. LINE 210
“…the classification accuracy …“ should be “…the classification accuracy (CA)….”
11. LINE 218
“K was 5 in … “ is not a correct sentence.
12. LINE 225
“…shows …“ should be “…showed….”
13. LINE 252
“…feature of …“ should be “…features of ….”
14. LINE 253
“…higher 80%.“ should be “…higher than 80%.”
15. LINES 261~262
When the motion types are mentioned in a sentence, replace motion type by ‘motion type’ for clearness.
16. LINES318~320
This sentence “In this paper, … is known.“ is not correct one. However, it can be divided into several sentences.
Because there are too many errors in spelling or writing in the mauscript, the reviewer strongly commends that the authors should carefully modify the whole context of the manuscript.
Author Response
Dear reviewer
Thanks for the reviewer’s advice.
To correct the language errors in the draft, we have made the modification according to the reviewer’s advice and three of the authors have checked and revised the paper carefully.
The modified draft could be seen in the attachment.
Yours sincerely, Yue Zhang
